# The Management of Cholestatic Liver Diseases: Current Therapies and Emerging New Possibilities

**DOI:** 10.3390/jcm10081763

**Published:** 2021-04-18

**Authors:** Marta Mazzetti, Giulia Marconi, Martina Mancinelli, Antonio Benedetti, Marco Marzioni, Luca Maroni

**Affiliations:** 1Clinic of Gastroenterology and Hepatology, Università Politecnica delle Marche, 60126 Ancona, Italy; giuliamarconi90@gmail.com (G.M.); martinamanci@gmail.com (M.M.); antonio.benedetti@ospedaliriuniti.marche.it (A.B.); m.marzioni@staff.univpm.it (M.M.); luca.maroni@live.it (L.M.); 2Department of Gastroenterology, Azienda Sanitaria Unica Regionale Marche Area Vasta 3, 62100 Macerata, Italy

**Keywords:** primary biliary cholangitis (PBC), primary sclerosing cholangitis (PSC), clinical trials, ursodeoxycholic acid (UDCA), Farnesoid X Receptor (FXR) agonist, Pan-Peroxisome Proliferator-Activated Receptor (PPAR) agonists

## Abstract

Primary biliary cholangitis (PBC) and primary sclerosing cholangitis (PSC) are two chronic cholestatic liver diseases affecting bile ducts that may progress to biliary cirrhosis. In the past few years, the increasing knowledge in the pathogenesis of both diseases led to a growing number of clinical trials and possible new targets for therapy. In this review, we provide an update on the treatments in clinical use and summarize the new drugs in trials for PBC and PSC patients. Farnesoid X Receptor (FXR) agonists and Pan-Peroxisome Proliferator-Activated Receptor (PPAR) agonists are the most promising agents and have shown promising results in both PBC and PSC. Fibroblast Growth Factor 19 (FGF19) analogues also showed good results, especially in PBC, while, although PBC and PSC are autoimmune diseases, immunosuppressive drugs had disappointing effects. Since the gut microbiome could have a potential role in the pathogenesis of PSC, recent research focused on molecules that could change the microbiome, with good results. The near future of the medical management of these diseases may include new treatments or a combination of multiple drugs targeting different signaling pathways at different stages of the diseases.

## 1. Introduction

Primary biliary cholangitis (PBC) and primary sclerosing cholangitis (PSC) are two chronic inflammatory autoimmune diseases of the bile ducts, which could culminate in biliary cirrhosis. Very few treatment options were available for decades, but in the past years many new targets and therapies were investigated, and clinical trials were performed.

The aim of this review is to provide an update on new targets and novel therapies that may change the management of these diseases in the near future.

## 2. Primary Biliary Cholangitis

PBC is a chronic autoimmune cholestatic liver disease that predominantly affects women. It is characterized by cholestasis, serologic reactivity to antimitochondrial antibodies (AMA) or to specific antinuclear antibodies (ANA) such as Sp100 and Gp210, and histologic evidence of chronic non-suppurative, granulomatous, lymphocytic small bile duct cholangitis. Many aspects of the aetiology and the pathogenesis of the disease are still uncertain, and the disease is often progressive, resulting in chronic cholestasis and possibly cirrhosis [1,2]. The main treatment goals include the prevention of the progression of the disease and the management of the symptoms, which may have a strong negative impact on the quality of life of patients. The only two medications approved by the Food and Drug Administration (FDA) are ursodeoxycholic acid (UDCA) and obeticholic acid (OCA). However, over the past years, given the strong support of randomized clinical studies, new therapies entered into the clinical practice of many experts in the field. Moreover, others molecules are actively being investigated in different clinical trials with promising results [3]. In this section, we are going to review the principal drugs in clinical use, in clinical trial, an in a preclinical phase for PBC.

### 2.1. Therapies in Clinical Use

#### 2.1.1. UDCA

UDCA, at a dosage of 13–15 mg/kg/day, is the first-line treatment for PBC [1]. It is the 7-β epimer of the chenodeoxycholic acid, a human bile acid. The complex mechanisms of action of UDCA and the evidence for its clinical use are extensively reviewed elsewhere [2,4]. Several molecular mechanisms contribute to the beneficial effect of UDCA in PBC patients. Indeed, many studies have shown that UDCA has anti-cholestatic effects due to complex post-transcriptional molecular mechanisms, a cytoprotective property, thanks to its action on endoplasmic reticulum stress, and an anti-inflammatory activity, inhibiting prostaglandin E2 [5]. UDCA administration also makes the endogenous bile acid pool more hydrophilic, and it improves therefore the biliary bicarbonate (HCO3^−^) umbrella, which is thought to create a protective layer on the apical surface of cholangiocytes against the permeation of protonated bile acids [6]. Moreover, UDCA interferes with the pathogenesis of autoimmune diseases by decreasing the expression of Major Histocompatibility Complex (MHC) class I and class II, the eosinophil levels in blood, and the immune reaction against PAMPs [7]. The administration of UDCA in PBC patients induces a reduction in markers of cholestasis, IgM, and AMA level [8]; improves liver histology [9]; and decreases mortality, especially when started at early stage [10]. Unfortunately, one-third of the patients have an inadequate response to UDCA treatment, defined according to several scoring systems, including the Barcelona, Paris I, Paris II, Rotterdam, Toronto, Ehime, GLOBE, and UK-PBC scoring systems [1]. Recently, the UDCA Response Score (URS), calculated with pre-treatment parameters, was used to predict the UDCA response [11]. A lower probability of UDCA response was significantly associated with a higher level of ALP (*p* < 0.0001), higher levels of total bilirubin (*p* = 0.0003), lower aminotransferase concentration (*p* = 0.0012), younger age (*p* < 0.0001), longer gap from diagnosis to UDCA treatment (*p* < 0.0001), and worsening of ALP from diagnosis (*p* < 0.0001). Based on these variables, the score reached an area under the receiver operating characteristic curve of 0.83 in predicting UDCA response. Other factors that contribute to the response to treatment are male sex [12], PBC-specific ANA positivity [1], and histology [11].

#### 2.1.2. Steroidal FXR Agonist: Obethicolic Acid (OCA)

OCA is an analogue of chenodeoxycholic acid (CDCA), with the addition of an ethyl group which gives a strong affinity for the nuclear farnesoid X receptor (FXR). FXR is the primary regulator of bile acid homeostasis, thanks to its effect on reducing production and reabsorption and increasing excretion [13]. After the good results of two phase II studies and one phase III clinical trial (POISE), in October 2016, OCA reached the EMA authorization for PBC treatment. The POISE study was a 12-month, double-blind, randomized, placebo-controlled phase III trial, evaluating 216 patients. The study included three treatment arms: OCA 10 mg ± UDCA, titration arm (OCA 5 mg ± UDCA for six months and then OCA 10 mg for the following six months), and placebo ± UDCA. The primary endpoint (i.e., ALP < 1.67 together with ALP reduction of at least 15% from baseline and normalization in total bilirubin) was reached by 46% and 47% of patients in the 5–10 mg and 10 mg OCA arms, respectively, and by 10% in the placebo group. Treatment arms also had a reduction in ALP, AST, and GGT that reached their lowest levels after three months of treatment and were maintained up to 48 months. The main adverse event was pruritus, which caused the study interruption for 7 out of 73 patients in the OCA 10 mg group, and in 1 out of 70 in the titration arm. Concerning the lipid profile, a transient increase in LDL and a decrease in HDL, VLDL, total cholesterol, and triglycerides were detected [14,15]. The long-term efficacy and safety of OCA for PBC patients who are intolerant to UDCA or have an inadequate response to UDCA were confirmed in the three-year interim analysis of the five-year open-label extension of the pivotal phase 3 POISE trial [16]. Moreover, a sub-analysis of data from the POISE study showed that OCA treatment was associated with improvement or stabilization of histological features of the disease (ductular injury, fibrosis, and collagen deposition), but final analyses of fibrosis-related endpoints are ongoing [17]. OCA monotherapy (10 mg and 50 mg) was also studied in a double-blind, placebo-controlled phase 2 study in patients with PBC. After three months, a significant decrease in ALP was observed in both of the groups, and a similar effect was detected through six years of open-label extension treatment [18]. Thus, OCA is recommended by international guidelines as a first-line therapy in patients who are intolerant to UDCA, and as a second-line therapy in addition to UDCA in patients with an incomplete response to UDCA. Of note, special attention should be paid in cirrhotic patients. In fact, severe liver injury or death was reported in patients treated with incorrectly high doses, and the FDA has issued a Black Box Warning for OCA. Guidelines recommend starting OCA at a dose of 5 mg weekly (with a maximum dose of 10 mg twice weekly) in Child Pugh B or C cirrhotic patients, and to use caution in Child Pugh A patients [1,19,20].

#### 2.1.3. PPARs Agonist: Bezafibrate

Bezafibrate is a pan-peroxisome proliferator-activated receptor (PPAR) agonist and, in combination with UDCA, was demonstrated to have a potent activity in PBC due to its specific anticholestatic properties. PPARs are nuclear receptors regulating the transcription of genes involved in metabolic pathways and inflammation. They exist in three isotypes (PPAR-α, PPAR-γ, and PPAR-β/δ), with different tissue distributions and actions. PPARα are mainly expressed in hepatocytes, where they stimulate multidrug resistance protein 3 (MDR3) expression, which protects cholangiocytes against bile salt due to its effect on phosphatidylcholine secretion [21]. Moreover, PPARα has an anti-inflammatory action that is based on trans-repression of AP1 and NF-kB signaling, transcription factors responsible for the expression of many genes involved in inflammation, oncogenesis, and apoptosis [2]. PPARβ/δ, specifically expressed in hepatocytes, cholangiocytes, Kupffer cells, and hepatic stellate cells, plays a role in the progression of PBC due to its anti-inflammatory effects. PPARδ is also involved in the transport and the absorption of bile components [22]. PPAR-γ, expressed in Kupffer cells, has anti-inflammatory activity, and its agonist is proved to reduce portal inflammation in murine models of PBC [23]. Bezafibrate was evaluated in the BEZURSO trial, a two-month, double-blind, randomized, placebo-controlled phase 3 trial, in which the combination of UDCA and bezafibrate 400 mg was compared with UDCA and placebo in 100 patients who had an inadequate response to UDCA according to the Paris 2 criteria. The primary endpoint of the study was a complete biochemical normalization at 24 months. Interesting, the primary endpoint was achieved by 37% of patients treated with bezafibrate and 0% of patients in the control group. Moreover, 67% of the patients treated with bezafibrate reported a normalization of ALP, compared to 2% in the placebo group. Itch improved in almost one-third of patients. Histologic data were too limited to determine whether bezafibrate had a role in the reduction of liver fibrosis and hepatic inflammation; however, a significant decrease in liver stiffness and Enhanced Liver Fibrosis score was observed. With the exception of the well-known side effects of fibrates (myalgias and increases in creatinine and transaminases), no statistical differences regarding adverse events between the two groups were observed. As a precaution, bezafibrate should be administered with caution in patients at risk for chronic kidney disease (e.g., diabetes, hypertension, or established renal disease) [24]. Moreover, another study on PBC patients with a suboptimal response to UDCA proved that a long-term treatment with UDCA and bezafibrate has an excellent effect on pruritus. As a matter of fact, after a median of 38 months, all but one patient reported a partial or complete itching relief, and a recurrence or worsening of pruritus was observed after bezafibrate discontinuation [25].

Fenofibrate is another PPARα-agonist, and it was also studied in PBC patients. A retrospective study on patients treated with UDCA and fenofibrate, compared with patients treated only with UDCA, proved that the fenofibrate-treated group had a significant improvement in the biochemical parameters, in particular ALP and ALT [26]. The same effect on ALP was demonstrated in another retrospective study on PBC patients with a suboptimal response to UDCA treated with fenofibrate and UDCA [27], but more studies and randomized controlled trials are needed to understand its role in PBC.

#### 2.1.4. Corticosteroid: Budesonide

Budesonide is a potent synthetic corticosteroid with a high first-pass metabolism within the liver, resulting in few systemic side effects compared to other systemic steroids. It is an agonist of the nuclear glucocorticoid receptor (GR) and pregnane X receptor (PXR). Budesonide and UDCA have a synergic activity in increasing the expression of the biliary chloride/bicarbonate anion exchanger 2 (AE2) with the result of an increase in biliary secretion of bicarbonate and stabilization of the biliary bicarbonate umbrella [3]. Previous studies showed that budesonide improves liver histology and biochemistry in PBC patients with interface hepatitis on biopsy [28,29]. In contrast, in a recent three-year phase-III, double-blind, randomized trial comparing budesonide vs. placebo, patients treated with UDCA showed that budesonide combined with UDCA was not associated with an improvement in liver histology in patients with PBC and an inadequate response to UDCA. It is important to mention that the study was underpowered for the evaluation of the liver histology due to challenges in patient recruitment. Improvements in biochemical markers of disease activity were demonstrated in secondary analyses [30]. Budesonide should be avoided in cirrhotic patients because of the increased risk of portal vein thrombosis and uncontrolled systemic shunting of the drug [31].

### 2.2. Therapies Evaluated in Clinical Trials

The main aspects of the clinical trials are described in Table 1.

#### 2.2.1. Non-Bile Acids FXR Agonists

Many FXR non-steroid agonists were investigated in PBC.

Cilofexor, a synthetic nonsteroidal FXR ligand, is involved in the transcriptional regulation of genes that play a role in bile acid metabolism. Cilofexor was tested in a phase 2 placebo-controlled, 12-week study on PBC patients. Cilofexor 100 led to a decrease in ALP (median reduction −13.8%; *p* = 0.005 vs. placebo), in GGT (−47.7%; *p* < 0.001), in ALT (−17.8%, *p* = 0.08), and in C-reactive protein (CRP; −33.6%, *p* = 0.03). Unfortunately, grade 2–3 pruritus occurred in 39% of the patients treated with Cilofexor 100 mg, compared with 10% in Cilofexor 30 mg and in 8% of patients treated with placebo. Pruritus led also to treatment discontinuation in 7% of patients on Cilofexor 100 mg [32].

Tropifexor (LJN452) is a non-bile acid FXR agonist investigated in a double-blind, randomized, placebo-controlled, phase 2 study (“A Multi-part, Double Blind Study to Assess Safety, Tolerability and Efficacy of Tropifexor (LJN452) in PBC Patients”, NCT02516605) that evaluated the safety and the efficacy of different doses of Tropifexor (30 µg, 60 µg, and 90 µg) in patients with an inadequate response to UDCA [33]. As opposed to OCA, Tropifexor should not have major effects on the lipid profile, being a non-steroidal molecule. To elude the confounding effect of ALP gene induction mediated by FXR, the endpoint of this trial was set on the reduction in GGT levels. After four weeks, interim analysis showed a dose-dependent reduction in GGT, ALP, and hepatocellular damage (ALT). Therefore, this study indicates the potential benefit of Tropifexor in PBC, and further studies are warranted [45].

EDP-305 is another FXR agonist that was evaluated in PBC because of its antifibrotic effect in animal models [46]. A phase 2 double-blind, placebo-controlled trial assessing the safety, pharmacokinetics, and efficacy in patients with PBC and inadequate response or intolerance to UDCA was just completed (“A Study to Assess the Safety, Tolerability, Pharmacokinetics and Efficacy of EDP-305 in Subjects With Primary Biliary Cholangitis”, NCT03394924). In the intent-to-treat analysis recently announced, EDP-305 did not meet the primary endpoint as defined by at least a 20% reduction in ALP, but key secondary endpoints (changes in ALT, AST, and GGT compared with placebo) at week 12 were reached in both the EDP-305 1 mg arm and the 2.5 mg arm.

#### 2.2.2. PPAR Agonists

Seladelpar is a new selective agonist of the PPARδ receptor, which has an anti-inflammatory and choleretic activity. The first phase 2 clinical trial that investigated the effect in PBC patients nonresponsive to UDCA was prematurely terminated because of the occurrence of a reversible grade 3 increase in transaminase levels in three patients [34]. A new phase 2 study evaluating a lower dose of Seladelpar (5 mg and 10 mg) was recently performed. The 12-week interim results, first published at the AASLD Liver Meeting in 2017, showed a drop in ALP in 45% and 82% of patients in the 5 mg group and 10 mg group, respectively, and a normalization of ALP in 12% of the 5 mg group and 45% of the 10 mg group, respectively [35]. Given the promising results of the interim analysis, another clinical trial evaluating the efficacy and the safety of Seladelpar 2 mg, 5 mg, and 10 mg is ongoing (NCT02955602). Finally, at the end of 2018, the ENHANCE trial started. It was a 52-week, double-blind, placebo-controlled, randomized phase 3 study that included subjects with PBC and an inadequate response to UDCA or intolerance to UDCA (“ENHANCE: Seladelpar in Subjects With Primary Biliary Cholangitis (PBC) and an Inadequate Response to or an Intolerance to Ursodeoxycholic Acid (UDCA)”, NCT03602560) [45]. Unfortunately, the open-label extension phase of this study was suspended after the onset of a similar trial evaluating the role of Seladelpar in NASH that found the occurrence of interface hepatitis in histological specimens. However, an independent panel of expert hepatologists and pathologists deemed that study-stopping was not warranted, since liver injury was within the expected changes seen in NASH patients and could not be attributed to Seladelpar. Recruitment has therefore restarted for Seadelpar in PBC patients after being put on hold. The phase 3 RESPONSE trial (NCT04620733) is currently recruiting patients.

Elafibranor, a dual PPAR-α/δ agonist, also studied in non-alcoholic steatohepatitis (NASH) [47], was recently tested in a multicenter, randomized, double-blind, placebo-controlled phase 2 study clinical trial recruiting patients with PBC non-responders to UDCA. Data were discussed at the International Liver Congress in Vienna in April 2019 [36]. Forty-five patients were randomized into three arms: Elafibranor 80 mg, Elafibranor 120 mg, and placebo. After 12 weeks of treatment, a reduction in ALP from baseline was observed in 48% patients in the 80 mg group and in 41% in the 120 mg arm; an increase of 3% was detected with placebo. Moreover, 67% patients in the 80 mg group (*p* = 0.001) and 79% of patients in the 120 mg group (*p* < 0.001) reached the secondary endpoint (serum ALP < 1.67 ULN, ALP decrease > 15%, total bilirubin < ULN) (NCT03124108). Thus, in July 2019, the USA FDA and the European Medicines Agency approved Orphan Drug Designation to Elafibranor for the treatment of PBC [48].

#### 2.2.3. Fibroblast Growth Factor 19 (FGF19) Analogues

FGF19 acts as a hormone on a cell surface receptor complex in hepatocytes, decreasing bile acid synthesis, gluconeogenesis, and lipogenesis. FGF19 expression is induced by bile-acid-mediated activation of FXR in the gut [49], and it reaches the liver through portal circulation. In the liver, FGF19 suppresses bile acid synthesis due to the inhibition of cholesterol 7-α-hydroxylase (CYP7A1) and sterol 12-α-hydroxylase (CYP8B1). Moreover, FXR decreases hepatic fibrogenesis by reducing collagen and by increasing matrix metalloprotease activity in hepatic stellate cells [50].

NGM282 (Aldafermin), an engineered analogue of FGF19, was tested in a 28-day, double-blind, placebo-controlled phase 2 trial. Forty-five PBC patients with an inadequate response to UDCA were treated with subcutaneous daily doses of NGM282 at 0.3 mg (n = 14), 3 mg (n = 16), or placebo (n = 15). ALP level had a significant drop in the treatment group, as well as transaminase levels and markers of cholestasis, hepatocellular injury, and inflammation (IgM levels). The reduction in complement component 4 (C4) levels suggests that NGM282 acts with a direct inhibition in the de-novo bile acid synthesis through the classical pathway. The main adverse effect was diarrhea. No effect on itch was detected [37]. In contrast to FGF19, no increase in liver cancer risk was observed in animal models treated with NGM282 [51]. Longer studies are needed to evaluate the long-term efficacy and safety of this molecule.

#### 2.2.4. Antifibrotic Agent

Setanaxib (GKT137831) is an inhibitor of Nicotinamide Adenine Dinucleotide Phosphate (NADPH) oxidases isoforms 1 and 4. NADPH oxidase enzymes, generating reactive species of oxygen, play a central role in inflammation and stellate cell-mediated fibrogenesis [52]. It was demonstrated in animal models of acute biliary injury and steatohepatitis that GKT137831 reduces hepatocyte apoptosis and liver fibrosis [53]. Thus, a multicenter, randomized, double-blind, placebo-controlled phase 2 study evaluating the safety and the efficacy of GKT137831 OD or BID in 111 patients with PBC and incomplete response to UDCA was performed (NCT03226067). Interim analysis showed a reduction in GGT and ALP level in six weeks, without a significant concomitant adverse event. A decrease in GGT of 7%, 12%, and 23% were observed in the placebo, 400 mg OD, and 400 mg BID groups, respectively (*p* < 0.01 for 400 mg BID vs. placebo). A greater GGT reduction was reached in patients with more advanced disease (GGT ≥ 2.5 X ULN at baseline). Changes in ALP were statistically significant in the 400 mg BID versus placebo [38].

#### 2.2.5. Immunomodulatory Strategies

Since PBC is an autoimmune condition characterized by anti-mitochondrial autoantibodies (AMA) and high levels of immunoglobulin M (IgM), many immunosuppressive drugs were studied in PBC, including corticosteroid [54], azathioprine [55], cyclosporine [56], methotrexate [57], and mycophenolate mofetil [58]. However, results were consistently unsatisfactory. Recently, other molecules were studied in PBC.

Rituximab, an anti-CD20 antibody currently used in lymphomas and autoimmune syndromes, was evaluated in PBC due to its promising results in murine models of autoimmune cholangitis [58]. Three clinical trials in PBC patients with an incomplete response to UDCA were reported. In an open label study, Rituximab (two doses of 1000 mg) induced a decrease in AMA and IgM levels, with only a marginal reduction of ALP after 36 weeks [39]. Unfortunately, a similar study including 14 PBC patients showed a significant but only transitory reduction in ALP [40]. Finally, Rituximab was demonstrated not to have an impact on fatigue, assessed by PBC-40 [41].

Ustekinumab is an anti-interleukin (IL)-12/23 monoclonal antibody commonly used in several autoimmune syndromes and inflammatory bowel diseases (IBD). IL-12 and IL-23-mediated Th1/Th17 signaling pathways play a role in the etiopathogenesis of PBC [59]. Unfortunately, a multicenter open label trial did not reach the primary endpoint of reduction in ALP of 40% from the baseline. However, at week 24, a statistically significant decrease of 12.1% in ALP from baseline was observed [42].

Abatacept is a Cytotoxic T-Lymphocyte Antigen 4 IgG antibody used in rheumatoid and psoriatic arthritis. An open-label, 24-week trial was performed in PBC patients, but no significant changes in biochemical enzymes were observed [43].

The efficacy of Baricitinib (LY3009104), a reversible inhibitor of Janus kinase 1 (JAK1) and JAK2 currently used in rheumatoid arthritis, is currently being evaluated in an ongoing, placebo controlled phase 2 trial (NCT03742973) [45].

Other types of molecules are undergoing clinical evaluation in phase 1 and phase 2 trials: FFP104 blocks the CD40/CD40L interaction between CD4+ T helper lymphocytes and B cells that are involved in the pathogenesis of PBC (NCT02193360) [60]; E6011 is an anti-chemokine-adhesion molecule CX3CL1 (fractalkine) antibody, which is elevated in the serum of PBC patients (NCT03092765); Etrasimod is a selective sphingosine-1-phosphate (S1P) receptor (S1PR) modulator targeting S1P receptor subtypes 1, 4, and 5, leading to an inhibition of activated lymphocytes from migrating to sites of inflammation (NCT03155932) [3].

#### 2.2.6. Other Treatment

S-adenosyl-L-methionine, added to UDCA in non-cirrhotic PBC patients, was demonstrated to have a positive effect on markers of cholestasis and quality of life, probably due to its hepatoprotective effects [44]. In this open label on 24 PBC patients, there was a significant decrease of ALP, GGT, and total cholesterol over a period of six months. A significant improvement of fatigue and pruritus on the PBC-40 questionnaire was also observed.

### 2.3. Therapies Evaluated in Pre-Clinical Studies

24-norursodeoxycholic acid (norUDCA) differs from UDCA due to the resistance in conjugation with taurine or glycine. NorUDCA increases the cholehepatic shunt of bile salts, leading to a supra-physiological secretion of bicarbonate. NorUDCA showed promising results in the treatment of PSC [61], but its efficacy in PBC has yet to be clarified. Up to now, improvements in fibrosis and inflammation were demonstrated in preclinical studies on animal model with cholestatic liver diseases [2].

Na+ -Taurocholate Cotransporting Polypeptide (NTCP) is a hepatocellular uptake transporter of bile salts, and its inhibition by myrcludex B results in hepatoprotective effects, increasing the biliary phospholipid/bile salt ratio. In 3.5-diethoxycarbonyl-1.4-dihydrocollidine-fed mice, a murine model of cholestasis, and in Atp8b1-G308V mice, used for chronic cholestasis, bile salt levels increased in treated animals from 604 ± 277 to 1746 ± 719 μm and from 432 ± 280 to 762 ± 288 μm, respectively, while phospholipid output was maintained, resulting in a higher phospholipid/bile salt ratio. Thus, it may be beneficial in some forms of cholestasis, but further studies need to be performed [62].

## 3. Primary Sclerosing Cholangitis

Primary sclerosing cholangitis (PSC) is a chronic bile duct disease with a prevalence of 1–16 per 100,000. PSC is more common in men (comprising 60–70% of patients) and is reported more frequently in Northern European countries and in North America. Moreover, 70% of the patients have ulcerative colitis [63]. The diagnosis is based on a combination of clinical, laboratory, imaging, and histological factors. Endoscopic retrograde cholangiopancreatography (ERCP) plays a very limited role in the diagnosis of PSC, while it may be used for the treatment of dominant stenosis [64]. It is well-known that patients affected by PSC have a higher risk of cholangiocarcinoma and gallbladder cancer. Up to now, no pharmacological treatment is universally approved for PSC. The lack of a clear pathogenesis and the absence of consistent endpoints have contributed to the difficulties in unravelling novel molecular targets and in designing effective clinical trials for PSC treatment [45]. The principal promising treatments and ongoing trials will be summarized in this section.

### 3.1. Therapies in Clinical Use

#### UDCA

The use of UDCA in PSC patients remains controversial to date. Previous small and uncontrolled studies of short duration consistently reported an improvement in liver tests in PSC treated with UDCA [65,66]. The first randomized controlled trial of UDCA (13 to 15 mg/kg) in PSC patients appeared in 1992. Beuers et al. showed a significant improvement of biochemical parameters, such as bilirubin, ALP, GGT, and transaminases, in six PSC patients treated for one year as compared to placebo [67]. A number of subsequent studies evaluated the effect of UDCA at different dosages in PSC. Despite the amelioration of biochemical parameters that appears to be relatively constant in all studies, definite proof for an improvement in “hard endpoints” such as survival, liver transplantation, or progression to CCA is still lacking. In a small cohort of 26 PSC patients, Mitchell et al. reported beneficial effects of UDCA (20 mg/kg) not only on liver tests but also on the cholagiographic appearance of the biliary tree evaluated by ERCP and liver fibrosis [68]. A subsequent randomized controlled trial in 219 PSC patients treated with UDCA (17 to 23 mg/kg) or placebo failed to show a significant improvement in the combined endpoint “death or liver transplantation”, despite a trend to a reduction in both (31% and 34% reduction, respectively) [69]. Moreover, high doses of UDCA in the range of 28–30 mg/kg were shown to be associated with an increased risk of disease progression to cirrhosis, development of varices, CCA, liver transplantation, or death [70]. Unfortunately, three meta-analyses also failed to show an effect of UDCA on mortality or liver transplantation [71,72,73]. To date, the most recent guidelines by the British Society of Gastroenterology recommend not to treat newly diagnosed PSC patients with UDCA routinely [74].

### 3.2. Therapies Evaluated in Clinical Trials

The principal characteristics of the clinical trials are described in Table 2.

#### 3.2.1. 24-Norursodeoxycholic Acid (norUDCA)

24-norursodeoxycholic acid (norUDCA) has a molecular structure similar to UDCA, except for the lack of a methylene group, resulting in a resistance to conjugation. NorUDCA is therefore passively absorbed from cholangiocytes and goes through cholehepatic shunt, which leads to the stimulation of a bicarbonate-rich choleresis. Moreover, norUDCA has anti-lipotoxic, anti-proliferative, anti-fibrotic, and anti-inflammatory effects, and, in vitro, it is less toxic than UDCA for hepatocytes and cholangiocytes due to its hydrophilicity [2]. A phase 2 clinical trial on 161 PSC patients without concomitant UDCA therapy, evaluating the efficacy of three doses of oral norUDCA, showed a significant dose-dependent reduction in ALP values after 12 weeks, without significant adverse events. The authors showed a significant reduction in ALP levels of 12.3%, 17.3%, and 26.0% in patients treated with 500 mg, 1000 mg, and 1500 mg per day of norUDCA, respectively; placebo-treated patients had a minor increase in ALP levels (1.2%) [61]. Despite some concerns of possible worsening of the disease due to the choleretic effects of norUDCA (especially in PSC patients with dominant strictures), these effects need to be clarified in longer studies; the association of UDCA and norUDCA has the potential to offer additive beneficial effects for PSC patients [87]. A phase 3 double-blind, randomized clinical trial is actively recruiting patients across several worldwide centers (NCT03872921).

#### 3.2.2. FXR Agonists

FXR agonists are evaluated in PSC because of their inhibition in bile acid synthesis in the liver, as previous explained [45].

OCA was tested in PSC patients in the AESOP trial (a randomized, double-blind, placebo-controlled phase II study). Seventy-seven PSC patients were recruited, and they were treated for 24 weeks with titrating doses of 1.5–3 mg/day and 5–10 mg/day OCA, or placebo, after 12 weeks. At the end of the study, serum ALP was significantly reduced with OCA 5–10 mg compared with the placebo arm (least-square mean difference of −83.4 U/L; *p* = 0.043). Interestingly, the effective dose of OCA is already in use for PBC therapy. The effect of OCA 5–10 mg was independent of administration of UDCA, despite a greater reduction in ALP that was registered in patients without UDCA at baseline (25–30% ALP reduction in patients without UDCA at baseline vs. 14–16% ALP reduction in patients with UDCA at baseline). The main side effect was dose-dependent pruritus, which occurred in 67% of patients in the OCA 5–10 mg group, in 60% of patients in the OCA 1.5–3 mg group, and in 45% of patients in the placebo arm. Discontinuation due to pruritus occurred only in one patient in the OCA 1.5–3.0 mg group and in three patients in the OCA 5–10 mg group [75]. A phase 3 trial is actively recruiting patients (NCT02177136).

Cilofexor (GS-9674), a non-steroidal FXR agonist, was tested in a phase 2, randomized, double-blind, placebo-controlled trial of 52 non-cirrhotic PSC patients with ALP levels greater than 1.67 ULN. Patients treated with Cilofexor 100 mg had a significant drop in ALP, gamma-GT, ALT, and primary bile acids (ALP mean reduction of −13.8%, *p* = 0.005; gamma-GT mean reduction of 47.7%, *p* < 0.001; ALT mean reduction of −17.8%, *p* = 0.08; primary bile acids reduction of −30.5%, *p* = 0.0008). The main limitations of this trial were the inclusion of only large-duct PSC cases without cirrhosis and the low prevalence of IBD [76].

NGM282, a FGF19 analogue, was recently studied in a phase 2, randomized, double-blind, placebo-controlled trial in PSC patients with ALP levels greater than 1.5 × ULN. Despite that no significant changes in ALP from baseline were observed, fibrosis biomarkers (Enhanced Liver Fibrosis test score and Pro-C3) were significantly improved in the treatment group [77]. This trial has stimulated discussion about the most appropriated target in PSC [88]. There are no established endpoints in PSC. A recent consensus of the International Primary Sclerosing Cholangitis Study Group, reviewing available literature, concluded that the only few candidates as surrogate endpoints in PSC may be ALP, transient elastography, histology, or the combination of ALP and histology and bilirubin; however, no one exceeds level 3 validation [89].

All-trans retinoic acid (ATRA), currently used in acne and in acute promyelocytic leukemia, represses bile acid synthesis through the FXR/RXR nuclear receptor complex pathway [90]. The efficacy of the combination of UDCA (15–23 mg/kg/day) and ATRA (45 mg/m^2^/day) was tested in 15 PSC patients. Despite ATRA, admiration did not reach the primary endpoint of the study (30% reduction in serum ALP), and a decrease in ALT and C4 levels were observed [78]. An open-label phase 2 trial evaluating efficacy and the safety of a lower dose of ATRA is currently ongoing (NCT03359174).

#### 3.2.3. PPAR Agonists

There is a rising number of studies on the efficacy of fibrates in PSC. However, the majority of available data comes from observational or retrospective analyses [3]. A recent retrospective French-Spanish study reported a 40% reduction in ALP levels, together with amelioration of pruritus, after fenofibrate 200 mg/day or bezafibrate 400 mg/day treatment (median duration of therapy of about 1.5 years) in 20 PSC patients [91]. Interestingly, the authors reported a rebound in ALP levels after discontinuation of the PPAR agonist based on occurrence of biliary stones, tolerability, or worsening of liver tests. It has to be mentioned, however, that the liver stiffness evaluated by transient elastography significantly increased during the study. A small prospective study evaluated the efficacy of bezafibrate (200 mg bid) in 11 PSC patients. After 12 weeks of treatment, ALP and ALT levels significantly improved in 7 out of 11 (64%) patients and subsequently increased after treatment discontinuation [79]. Further studies on fibrates for PSC are warranted.

#### 3.2.4. Antifibrotic Therapy

Despite fibrosis being central in the pathogenesis of the disease, very few antifibrotic drugs have been studied. Lysyl oxidase like-2 (LOXL2) is an enzyme that catalyzes the crosslinking of collagen and elastin fibers, thereby strengthening the extracellular matrix structure. Previous studies showed that LOXL2 levels in the serum and liver of PSC patients are correlated with disease severity [92]. Moreover, the administration of a LOXL2 inhibitor in rodents was shown to reduce the accumulation of hepatic and biliary fibrosis and also accelerate its reversal [93,94]. Unfortunately, no improvement in liver fibrosis was observed in a placebo-controlled, phase 2b trial testing Simtuzumab, a LOXL2 inhibitor. In the trial, a total of 234 patients with compensated PSC were randomized on a 1:1:1 basis to receive placebo, weekly subcutaneous injections of Simtuzumab 75 mg, or weekly subcutaneous injections of Simtuzumab 125 mg for 96 weeks. The study failed to demonstrate any effect of Simtuzumab on hepatic collagen content (measured by morphometry on liver biopsy) and fibrosis stage (measured by the Ishak fibrosis stage) [80].

#### 3.2.5. Immunomodulators

Although PSC is an immune-mediated disease, traditional immunosuppressive approaches so far failed to demonstrate a clinical benefit in PSC [95]. Timolumab (BTT1023), a human monoclonal anti-VAP-1 antibody, was shown to prevent fibrosis in murine models of liver injury [96]. A phase 2 clinical trial (BUTEO trial) evaluating the effect of Timolumab in PSC over a 78-day treatment (primary endpoint: reduction of ALP levels by >25% from baseline) is still ongoing (NCT02239211) [97]. Cenicriviroc, a CCR2/CCR5 antagonist, was tested in a phase 2 trial (PERSEUS trial), and it was proven to cause a modest reduction in ALP (median 18%) after 24 weeks among 24 patients [98]. Moreover, it was shown to have anti-inflammatory and antifibrotic effects in NASH animal models and in Abcb4 (Mdr2−/−) mice [99].

Vedolizumab is a monoclonal antibody directed against the α4β7 integrin, which is used in the treatment of inflammatory bowel disease. MADCAM-1 is the ligand for α4β7 integrin and is normally expressed in the gut. Since MADCAM-1 is also found in the liver, administration of vedolizumab is thought to reduce MADCAM-1-induced leucocyte migration between the gut and the liver [100]. Despite these promising premises, in a retrospective analysis, Vedolizumab treatment did not show any improvement in liver biochemistry in patients affected by IBD and PSC who received at least three doses of vedolizumab. About 20% of patients experienced a reduction of at least 20% in ALP levels; however, this outcome was independently associated only with the presence of cirrhosis [81]. Similar results were reported in a previous retrospective study in 34 patients with PSC and IBD (16 patients affected by Crohn’s disease and 18 patients by ulcerative colitis) treated with vedolizumab [101].

Vidofludimus is an inhibitor of the dihydroorotate dehydrogenase that blocks the replication of activated T- and B-cells and interferes with the JAK/signal transducer [45]. A phase 2, open-label clinical trial evaluating the safety and the efficacy of vidofludimus in patients with PSC will start in 2020 (NCT03722576).

#### 3.2.6. Modulation of the Gut Microbiome

Recent research focused on the gut microbiome as a potential element in the pathogenesis of PSC. One of the hypothesis is that gut microbiome activates innate immunity within the liver, resulting in inflammation and fibrosis in the bile duct [102]. Moreover, studies on the microbiome and PSC demonstrated that the microbiome of PSC patients is different from healthy controls and IBD-patients [103]. Thus, changing the composition of the gut microbiome might reduce inflammation and fibrosis in the bile ducts.

Vancomycin is a glycopeptide antibiotic that also has an immunomodulatory effect due to the decrease in T cell cytokine production [104]. Vancomycin was compared to metronidazole [105] and to placebo [106] in two randomized trials in PSC patients with or without IBD, and a significant reduction in ALP levels and the Mayo score was reported. A phase 2, multicenter clinical trial aiming to recruit 102 adult participants with PSC and evaluating ALP levels at 6, 12, and 18 months is still ongoing (NCT03710122).

Other interesting antibiotics are rifaximin and minocycline. Rifaximin had no effect in decreasing cholestatic markers and the Mayo score in 16 PSC patients [82]. In contrast, minocycline was shown to cause an improvement in ALP levels and the Mayo score in 16 patients [83].

Fecal Microbiome Transplantation (FMT) is a promising treatment for PSC patients. In one small pilot study, patients with PSC underwent FMT, and three of them experienced a ≥50% decrease in ALP levels. Its effect may be correlated with the bacterial diversity and donor engraftment [84].

#### 3.2.7. Other Treatments

Anti-inflammatory drugs such as sulfasalazine and curcumin were tested in PSC patients. A multicenter, randomized, double-blinded, placebo-controlled trial to assess the benefit and the safety of sulfasalazine in the treatment of PSC just ended, and results are not available (NCT03561584). No significant improvements in cholestasis or symptoms were seen in patients treated with Curcumin [85].

Various minor drugs with different mechanisms of action could have a role in the treatment of PSC, and they were evaluated in different clinical trials. HTD1801 was studied in two phase 2 ongoing trials due to its action on lipid metabolism (NCT03333928, NCT03678480). DUR-928 is an endogenous epigenetic regulator that was studied in a phase 2 study on PSC patients due to its anti-inflammatory properties and its role in lipid metabolism and cell survival (NCT03394781) [3]. Docosahexaenoicacid supplementation, increasing PPAR signaling, was associated with a drop in ALP levels in patients with PSC, in a 12-month, open-label, pilot study on 23 PSC patients [86]. Another ongoing phase 1/2 trial is evaluating the potential effect of Hymecromone, a hyaluronic acid synthesis inhibitor (NCT02780752). Additionally, selected mesenchymal stromal cells (Orbcel-C) are in an ongoing phase 2 trial on PSC patients (NCT02997878) [3].

## 4. Current Therapeutic Management with Patients Newly Diagnosed with PBC and PSC

Overall, the current codified treatment for patients with PBC consists of UDCA, OCA, and bezafibrate. We provide a flowchart for the standard management of patients with a new diagnosis of PBC (Figure 1). Unfortunately, an analogue algorithm could not be performed for the management of PSC. As a matter of fact, as previously explained, there is not a codified treatment of PSC.

## 5. Conclusions

In this review, we provided an update on the drugs in clinical use and an overview of the new molecules in evaluation for the treatment of PBC and PSC patients. Recently, a deeper understanding of the pathophysiology of these diseases unveiled new molecular targets and, consequently, offered new chances for treatment. Given the complex nature of PBC and PSC, it appears unlikely that a single drug will be able to address all patients for each disease correctly. Instead, the near future of the medical management of chronic cholestatic liver diseases will most probably rely on a combination of multiple drugs targeting different signaling pathways at different stages of the disease. It will be essential to design clinical trials to address these issues specifically and to guide clinical management. A better knowledge of the molecular basis of the diseases and a more detailed disease stratification based on patient characteristics and disease behavior remain therefore the cornerstones to devise new effective treatments for PBC and PSC patients.

## Figures and Tables

**Figure 1 jcm-10-01763-f001:**
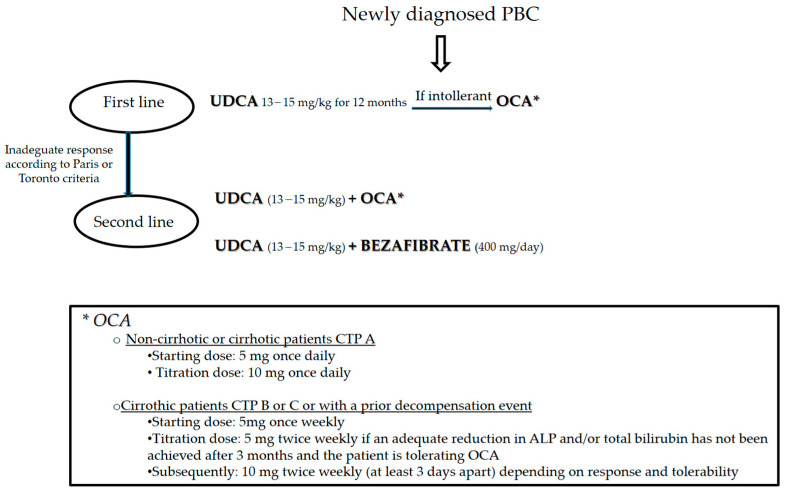
Current algorithm of the treatment in PBC.

**Table 1 jcm-10-01763-t001:** Principal characteristics of the study of the drugs in clinical trials.

	Study	Phase	Pt Number	Dose	Study Duration	Primary Endpoint	Primary Endpoint Met	Note
Non-Bile Acids FXR agonists (*drugs*)								
Cilofexor	[32]	2	71	30 mg, 100 mg	12 weeks	Safety and tolerability of Cilofexor	yes	
Tropifexor	[33]	2	61	30 µg, 60 µg, 90 µg	12 weeks	Change in GGT in 4 weeks	yes	at interim analysis
EDP-305	NCT03394924	2	68	Dose 1 dose 2	12 weeks	20% reduction in ALP or normalization of ALP in 12 weeks	n/a	ongoing
PPAR agonists (*drugs*)								
Seladelpar	[34]	2	70	50 mg, 200 mg	12 weeks	Change in ALP		Early stopped (grade 3 increases in ALT)
	[35]	2		5 mg, 10 mg	12 weeks	Change in ALP	yes	
	NCT02955602	2	119	2 mg, 5 mg, 10 mg	8 weeks with 44 weeks extension	Change in ALP	n/a	ongoing
	NCT03602560 (ENHANCE)	3	240 *	5–10 mg, 10 mg	52 weeks	Change in ALP and bilirubin		suspended (interface hepatits)
Elafibranor	[36]	2	45	80 mg, 120 mg	12 weeks	Change in ALP	yes	
Fibroblast growth factor 19 (FGF19) analogues (*drugs*)								
NGM282	[37]	2	45	0.3 mg, 3 mg	28 days	Change in ALP	yes	
Antifibrotic agent (*drugs*)								
Setanaxib	[38]	2	111	400 mg od/bd	24 weeks	Change in GGT	yes	at interim analysis
Immunomodulatory Strategies (*drugs*)								
Rituximab	[39]	Open label	6	1 g (2 doses)	52 weeks	Reduction in ALP, IgM and AMA after 36 week		
	[40]	Open label	14	1 g (2 doses)	6 months	Normalization or ALP < 25% from baseline	no	
	[41]	2	57	1 g (2 doses)	12 months	Fatigue (PBC 40)	no	
Ustekimumab	[42]	Open label	20	90 mg	28 weeks	ALP < 40% from baseline	no	
Abataceb	[43]	Open label	16	125 mg	24 weeks	ALP normalization or <40% from baseline	no	
Baricitinib	NCT03742973	2	2	2 mg, 4 mg	12 weeks	Change in ALP	no	Enrollment futility
FFP104	NCT02193360	1/2	24 (estimated)	1 mg/kg, 2.5 mg/kg, 2 mg/kg ev	12 weeks	Safety and tolerability	n/a	Recruitment status unknown
E6011	NCT03092765	2	29	High or low dose	64 weeks	ALP change at week 12	n/a	Terminated
Etrasimod	NCT03155932	Open label	2		24 weeks	ALP change	n/a	ongoing
Other treatment								
S-adenosyl-L-methionine	[44]	Open label	24	1.2 g	6 months	PBC 40 improvement	yes	significant decrease of ALP in non-cirrhotic patients

* estimated.

**Table 2 jcm-10-01763-t002:** Principal characteristics of the study of the drugs in clinical trials.

	Study	Phase	Pt Number	Dose	Study Duration	Primary Endpoint	Primary Endpoint Met	Note
24-norursodeoxycholic acid (norUDCA)	[61]	2	161	500 mg, 1 g, 1.5 gr	16 weeks	Change in ALP	yes	
	NCT03872921	3	300 *	250 mg *6 cps/d*	2 years	Change in ALP and histology	n/a	ongoing
FXR agonist (drugs)								
OCA	[75]	2	77	1.5–3 mg 5–10 mg	24 weeks	Change in ALP	yes	5–10 mg
Cilofexor	[76]	2	52	100 mg 30 mg	12 weeks	Safety and liver enzyme improvement	yes	
NGM282	[77]	2	62	1 mg 3 mg	12 weeks	Change in ALP	no	
ATRA	[78]	Pilot study	15	45 mg/m/d	12 weeks	ALP < 30% from baseline	no	Decrease in ALT and C4
	NCT03359174	2	2	10 mg bd	24 weeks	Change in ALP	n/a	ongoing
PPAR agonists								
Bezafibrates	[79]	2	11	200 mg BID	12 weeks	improvements in liver function test	yes	
Bezafibrates	[79]	2	11	200 mg BID	12 weeks	improvements in liver function test	yes	
Antifibrotic therapy (drugs)								
Simtuzumab	[80]	2	234	75 mg, 125 mg	96 weeks	Hepaticcollagencontent	no	
Immunomodulator (drugs)								
Timolumab	NCT02239211	2	23	8 mg/kg	11 weeks	ALP < 25% from baseline	n/a	Awaiting results
Cenicriviroc	NCT02653625	Open label	24	150 mg	24 weeks	Change in ALP	yes	
Vedolizumab	[81]	Retrospective	102		412 days (median)		no	ALP < 20% from baseline
Vidofludimus	NCT03722576	2	14	30 mg	6 months	Change in ALP	n/a	Awaiting results
Modulation of gut microbioma (drugs)								
Vancomycin	NCT03710122	2/3	102 *		24 months	Change in ALP	n/a	ongoing
Rifaximin	[82]	Open label	16	550 mg bd	12 weeks	Change in ALP	no	
Minocycline	[83]	Pilot study	16	100 mg bd	1 year	Change in biochemistry	yes	
FMT	[84]	Open label	10		24 weeks	safety	yes	
Other treatments (drugs)								
Sulfasalazine	NCT03561584	2	42	500 mg bd	14 weeks	Change in ALP	n/a	ongoing
Curcumin	[85]	Open label	258	750 mg bd	12 weeks	ALP < 1.5 ULN or <40% from baseline	no	
HTD1801	NCT03333928	2	59	500 mg 1 gr	18 weeks	Change in ALP	n/a	Awaiting results
DUR-928	NCT03394781	2	5	10 mg 50 mg	28 days	Change in ALP	n/a	ongoing
Docosahexaenoic acid	[86]	Open label	23	800 mg bd	12 months	Change in ALP and safety	yes	
Hymecromone	NCT02780752	1	18 *	1.2 gr 2.4 gr 3.6 gr	4 days	Change in spu	n/a	ongoing
Orbcel-C	NCT02997878	2	56 *	0.5, 1.0, 2.5 million cells/kg	56 days	Safety, change in ALP e ALT	n/a	ongoing

* estimated.

## Data Availability

No new data were created or analyzed in this study. Data sharing is not applicable to this article.

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
