# Peer review of "The Management of Cholestatic Liver Diseases: Current Therapies and Emerging New Possibilities"

_jcm, 2021, doi:10.3390/jcm10081763_

Round 1

Reviewer 1 Report

In general this is a nice review in a field that has seen some recent and ongoing therapeutic developments that warrant an update. My main concerns are minor and listed below:

  1. In the abstract I would state 'presumed' autoimmune disease for PSC. Also in line 15 there is an extra 'e'.
  2. Throughout the manuscript 'next future' is said - i believe 'near' is what was meant.
  3. lines 136138: this is a suggestion/precaution and should be phrased that way.
  4. For all of the tables - all of the reference numbering is incorrect and needs to be fixed.
  5. Table 1 - elafibranor is spelled incorrectly
  6. Table 1 - for SAMe in PBC (last row) there was also an improvement in serum ALP in the non-cirrhotic patients.
  7. Line 164-165: please remove 'has been proved to'
  8. Lines 203-206: the liver biopsy in the NAFLD study were deemed to not have warranted study stopping after expert panel review. The findings were within the spectrum of liver biopsy changes that can be seen with NASH. As a result of this review the study use of saladelpar in PBC has resumed after being put on hold.
  9. line 316: ERCP plays a very limited role in the diagnosis of PSC and should be avoided except for therapeutic needs (eg. dominant stricture) 
  10. Table 2 - all of the reference numbers are incorrect
  11. line 366: 'flushing effects' should be switched to something like 'choleretic' 
  12. 12. line 400: needs to be reworded
  13. line 403: 'revising' should be 'reviewing'
  14. line 449: should remove 'a count of'  Same for line 483-4
  15.  

Author Response

The authors wish to express gratitude to the Editor and the Reviewers for the consideration given to the manuscript entitled “The management of cholestatic liver diseases: current therapies and emerging new possibilities”. In particular, we thank the Reviewers for providing constructive criticisms, in order to improve the quality of our study. We have dealt with all the issues raised by the Reviewers and revised the manuscript according to their suggestions.

Below please find the point-by-point responses.

We thank the Reviewer for the attention given to our study; we took in consideration all her/his correction on how to improve the accuracy of our manuscript, as outlined below.

Point #1

In the abstract I would state 'presumed' autoimmune disease for PSC. Also in line 15 there is an extra 'e'.

Answer to Point #1

We agree with the Reviewer that the term autoimmune for PSC may be inappropriate. Since in the sentence we also refer to PBC, in order to simply the text, we preferred to delete it altogether. We are however willing to reformulate the whole sentence if deemed necessary. We thank the Reviewer for pointing out the mistaken “e” in the Abstract, which we have deleted.

Point#2

Throughout the manuscript 'next future' is said - i believe 'near' is what was meant.

Answer to Point #2

We acknowledge the mistake. We have replaced “next” with “near” throughout the test.

Point #3, 4, 5, 7, 9, 10, 11, 12, 13, 14

  1. lines 136138: this is a suggestion/precaution and should be phrased that way.
  2. For all of the tables - all of the reference numbering is incorrect and needs to be fixed
  3. Table 1 - elafibranor is spelled incorrectly
  4. Line 164-165: please remove 'has been proved
  5. line 316: ERCP plays a very limited role in the diagnosis of PSC and should be avoided except for therapeutic needs (eg. dominant stricture) 1
  6. Table 2 - all of the reference numbers are incorrect
  7. line 366: 'flushing effects' should be switched to something like 'choleretic' 
  8. line 400: needs to be reworded
  9. line 403: 'revising' should be 'reviewing'
  10. line 449: should remove 'a count of'  Same for line 483-4

Answer to Point #3, 4, 5, 7, 9, 10, 11, 12, 13, 14

We thank the Reviewer for the suggestions and for the corrections raised.  We have made all the adjustments.

Point #6

Table 1 - for SAMe in PBC (last row) there was also an improvement in serum ALP in the non-cirrhotic patients.

Answer to Point #6

The result of the study suggested by the Reviewer is very interesting and we put it in the note of the Table 1.

Point #8

Lines 203-206: the liver biopsy in the NAFLD study were deemed to not have warranted study stopping after expert panel review. The findings were within the spectrum of liver biopsy changes that can be seen with NASH. As a result of this review the study use of saladelpar in PBC has resumed after being put on hold.

Answer to Point #8

We thank the Reviewer for his/her correct and updated clarification. We have incorporated in the manuscript this concept.

Reviewer 2 Report

The paper authored by Mazzetti et al is a well-done comprehensive review on therapeutic strategies in PBS and PSC patients.

The topic of the study is very interesting, thus the paper is worth publishing.

I could only suggests some minor issues.

Regarding studies of OCA in PBC Patients, You could include the following:

(NCT00570765) Kowdley et al.  A randomized trial of obeticholic acid monotherapy in patients with primary biliary cholangitis. Hepatology. 2018 May; 67(5):1890-1902.

Regarding studies of Bezafibrate in PBC Patients, You could include the following:

Reig et al. Effects of bezafibrate on outcome and pruritus in primary biliary cholangitis with suboptimal ursodeoxycholic acid response. Am J Gastroenterol. 2018;113(1):49–55.

You could also shortly comment regarding the studies on fenofibrate in PBC ?

There is a type-setting error in the Table 2 regarding Docosahexanoid acid (DXA).

In the study of cilofexor in PSC (Trauner et al, Hepatology 2019) 52 patients were randomized.

Before the 4th paragraph (conclusions) You could describe (i.e. flowchart) the current therapuetic management with patient newly diagnosed with PBC, PSC.

Author Response

The authors wish to express gratitude to the Editor and the Reviewers for the consideration given to the manuscript entitled “The management of cholestatic liver diseases: current therapies and emerging new possibilities”. In particular, we thank the Reviewers for providing constructive criticisms, in order to improve the quality of our study. We have dealt with all the issues raised by the Reviewers and revised the manuscript according to their suggestions.

Below please find the point-by-point responses.

We thank the Reviewer for acknowledging the quality of the manuscript. We have dealt with all the useful comments of the Reviewer and largely adapted the manuscript to his/her suggestions.

 Point #1, 2, 3

  1. Regarding studies of OCA in PBC Patients, You could include the following:(NCT00570765) Kowdley et al. A randomized trial of obeticholic acid monotherapy in patients with primary biliary cholangitis. Hepatology. 2018 May; 67(5):1890-1902.
  2. Regarding studies of Bezafibrate in PBC Patients, You could include the following: Reig et al. Effects of bezafibrate on outcome and pruritus in primary biliary cholangitis with suboptimal ursodeoxycholic acid response. Am J Gastroenterol. 2018;113(1):49–55.
  3. You could also shortly comment regarding the studies on fenofibrate in PBC ?

Answer to Point #1, 2, 3

We thank the Reviewer for these important suggestions and for the interesting studies provided. We added them in the text.

Point #4, 5

There is a type-setting error in the Table 2 regarding Docosahexanoid acid (DXA).

In the study of cilofexor in PSC (Trauner et al, Hepatology 2019) 52 patients were randomized.

Answer to Point #4, 5

We thank the Reviewer for the corrections. We have changed them in the text.

Point #3

Before the 4th paragraph (conclusions) You could describe (i.e. flowchart) the current therapuetic management with patient newly diagnosed with PBC, PSC.

Answer to Point #3

The comment raised by the Reviewer is very pertinent and we added an algorithm for the explanation of the management of PBC. We preferred to comment in the text regarding the PSC algorithm, rather than presenting a picture, since PSC management is less uniform and often not completely supported by current literature.
